# Salicylic Acid-Induced Morpho-Physiological and Biochemical Changes Triggered Water Deficit Tolerance in *Syzygium cumini* L. Saplings

Zikria Zafar [1,2], Fahad Rasheed [1,*], Rana Muhammad Atif [3,4], Muhammad Maqsood [5] and Oliver Gailing [2,*]

1   Department of Forestry & Range Management, University of Agriculture, Faisalabad 38040, Pakistan; z.zafarfrw@gmail.com
2   Department of Forest Genetics and Forest Tree Breeding, University of Göttingen, 2 D-37077 Buesgenweg, Germany
3   Department of Plant Breeding and Genetics, University of Agriculture, Faisalabad 38040, Pakistan; dratif@uaf.edu.pk
4   Center of Advanced Studies in Agriculture and Food Security (CAS-AFS), University of Agriculture, Faisalabad 38040, Pakistan
5   Department of Agronomy, University of Agriculture, Faisalabad 38040, Pakistan; drmaqsoodbajwa@yahoo.com
*   Correspondence: fahad.rasheed@uaf.edu.pk (F.R.); ogailin@gwdg.de (O.G.)

**Abstract:** Fruit tree culture is at the brink of disaster in arid to semi-arid regions due to low water availability. A pot experiment was carried out to analyze whether foliar application of salicylic acid (SA) can improve water stress tolerance in *Syzygium cumini*. Saplings were subjected to control (CK, 90% of field capacity, FC), medium stress (MS, 60% of FC) and high stress (HS, 30% of FC) along with foliar application of 0.5 and 1.0 mM of SA. Results showed that soil water deficit significantly decreased leaf, stem and total dry weight, leaf gas exchange attributes and chlorophyll *a*, *b*. However, root dry weight and root/shoot ratio increased under MS and HS, respectively. Contrarily, foliar application of SA significantly improved chlorophyll *a*, *b*, leaf gas exchange attributes, and dry weight production under soil water deficit. Concentration of oxidants like hydrogen peroxide and superoxide radicals, along with malondialdehyde and electrolyte leakage increased under soil water deficit; however, decreased in plants sprayed with SA due to the increase in the concentration of antioxidant enzymes like superoxide dismutase, peroxidase, catalase and ascorbate peroxidase. Results suggest that the foliar application of SA can help improve water stress tolerance in *Syzygium cumini* saplings; however, validation of the results under field conditions is necessary.

**Keywords:** drought stress; Jaman; salicylic acid; assimilation rate; water use efficiency; osmolytes

## 1. Introduction

Increasing fossil fuel consumption, deforestation and industrialization is contributing towards a significant increase in global average air temperature, which has influenced the frequency and severity of drought events in many regions of the world [1,2]. Consequently, over the past decade, drought has become a significant global threat to plant survival and productivity [3]. It has been estimated that 36% of the global area falls under arid to semi-arid climate where annual precipitation is between 50–150 mm [4]. Under such a situation, most of the tree species are cultivated within a narrow hydraulic range where risk of hydraulic failure due to embolism has become common due to soil water deficit [5]. Pakistan falls under arid to semi-arid climate where annual precipitation is between 250–300 mm, and the country is facing a persistent problem of water shortage [6]. Moreover, being an agricultural country, the diversion of canal water towards farm crops has worsened the situation for fruit tree culture and tree plantations. Such a situation calls for urgent

measures that can ensure the survival and production sustainability of fruit trees under a changing climate that could rekindle the diminishing confidence of local farmers.

Water deficit has negative effects on plant growth that includes decrease in leaf size, plant height, and overall plant biomass. The noticeable reduction in plant development is driven by a decrease in plant turgor pressure, chlorophyll content and plant hormone balance due to water stress. Such changes induce variation at the cellular level and inhibit cell growth and development [7]. An increase in accumulation of proline and other osmolytes under water stress also helps sustain the cell turgor and allows cells to mitigate the harmful effects of water stress [8]. Moreover, water stress not only results in decreased stomatal conductance and $CO_2$ starvation under severe conditions [9], but also induces overproduction of reactive oxygen species (ROS) like superoxide radical ($O_2^-$), hydrogen peroxide ($H_2O_2$) and singlet oxygen $^1O_2$ in chloroplasts and mitochondria [10,11]. Plants scavenge the overproduction of the ROS by increasing the production of antioxidant enzymes like superoxide dismutase (SOD), peroxidase (POD), catalase (CAT) and ascorbate peroxidase (APX) [12]. Many studies reported that the balance between the production of ROS and antioxidant enzymes helps determine the plant tolerance to different types of stress [13].

Salicylic acid (SA) is a plant hormone involved in the regulation of plant growth and production and plants' response to different types of abiotic stresses [14]. Many studies have highlighted the role of SA in regulating physiological processes such as photosynthesis, osmolyte production, and antioxidant enzyme activity, thus improving the plant water–relation under stress [15]. Studies have observed that the application of SA either by seed soaking, mixing to the nutrient solution, or foliar application has resulted in significant increase in abiotic stress tolerance in plants [14]. The positive effects of SA are closely linked to different factors such as the species and developmental stage of plants, the mode of application, and the optimal concentration of SA [16]. Many studies have shown that application of SA helps maintain cell membrane stability [17] by increasing the concentration of various antioxidant enzymes (SOD, POD, CAT and APX) and improving the leaf photosynthetic capacity [18] under different abiotic stresses. An increase in leaf water potential and a decreased electrolyte leakage and lipid peroxidation have been evidenced after the application of SA [15]. The application of SA may induce or inhibit various plant functions under optimal or a high concentration of SA, respectively [14]. Therefore, although SA has been considered a short-term solution to improve plant tolerance to water stress [19], the optimal concentration of SA is species specific and remains unclear [20].

Jaman (*Syzygium cumini* L.), belonging to the family Myrtaceae [21,22], is a large tree with ovate and aromatic leaves. Young leaves are red/orange in color and the fruits are small, oval-shaped berries that turn dark purple to black in color on maturity [22]. Jaman is native to South Asia and has been introduced to Hawaii and across the Pacific for ornamental, timber and fruit purposes [21]. Jaman flourishes along stream banks and can tolerate seasonal flooding and moderate drought stress [22]. The leaves and fruits of Jaman are of high medicinal importance and are used to cure diabetics, chronic diarrhea and enteric disorders [23]. Jaman is growing in the irrigated and rain-fed regions of central and upper Punjab, Pakistan. In these regions, fruit trees undergo water stress especially during the growing season and under changing climate, and water stress has become a major problem for sapling survival and production sustainability of Jaman. The effect of SA application on water stress tolerance in *S. cumini* has never been elucidated, especially during early establishment stages. Therefore, this study aims to investigate whether SA application can effectively ameliorate water stress tolerance in *S. cumini* saplings. Various morphological, physiological and biochemical changes were recorded under water deficit and after application of SA in order to assess the tolerance status of plant saplings.

## 2. Materials and Methods

### 2.1. Plant Material and Growth Condition

A pot experiment was conducted in a greenhouse at the Department of Forestry and Range Management (31° 26′ N, 73° 06′), University of Agriculture, Faisalabad, Pakistan. Maximum and minimum temperature in the greenhouse was at 25 to 35 °C, relative humidity at 60%–70% and intensity of solar radiation was ~1200 photosynthetic photon flux density (PPFD). We procured 3- to 4-month-old healthy saplings of *S. cumini* from the Punjab Forest Research Institute Gatwala, Faisalabad, representing a single tree progeny. Plastic pots (34 cm diameter and 26 cm depth) were filled with sandy loam and farmyard manure (3:1 proportional) and total weight of plastic pots was maintained at 10 kg. The soil used in the experiment was analyzed for nitrogen (0.78%), phosphorus (12 ppm), organic matter (8%), electric conductivity (2 dS m$^{-1}$) and pH (6.6). To optimize the nutrient balance, NPK fertilizer (15% N, 5% $P_2O_5$, 5% $K_2O$) was added at a rate of 5 g/kg of soil.

### 2.2. Water Deficit Treatments and Foliar Application of Salicylic Acid (SA)

A total of 70 young, healthy and uniform-sized (18 ± 2 cm) saplings of *S. cumini* were allowed to grow under normal conditions. Ten saplings were randomly assigned to the following treatment combinations: CK, control at 90% field capacity; MS, medium stress at 60% field capacity; HS, high stress at 30% field capacity; MS + 0.5, medium stress sprayed with 0.5 mM SA; HS + 0.5, high stress sprayed with 0.5 mM SA; MS + 1.0, medium stress sprayed with 1.0 mM SA; HS + 1.0, high stress sprayed with 1.0 mM SA. The water deficit levels were maintained using methods as demonstrated by [24]. Every pot was watered back daily to the reference weight of pots adding the amount of water lost during evapotranspiration. Sodium salicylate (Merck, Darmstadt, Germany) was used to prepare the salicylic acid solutions (SA) by dissolving 0.069 g and 0.138 g of SA in 1 L of distilled water, respectively. During the experiment, plants under water stress were sprayed twice with 0.5 and 1.0 mM of SA (on the 7th and 45th day of the experiment). The plants under control, MS and HS were sprayed with distilled water and the experiment was continued for 90 days.

### 2.3. Growth and Dry Weight Production

Various growth parameters such as plant height (cm) stem diameter (mm), number of leaves were determined during the experiment. All the plants were uprooted at the end of the experiment and were divided into various plant sections (leaves, stems and roots) and fresh weigh was determined (G & G Electronic scale JJ3000B). All the plant sections were packed into paper bags and were oven dried for 72 h at 80 °C (DGH-9202 Series Thermal Electric Thermostat drying oven) to determine the dry weight of each plant section and total dry weight for each plant [25]. Root:shoot ratio (R:S ratio) was determined as the ratio of root dry weight and leaves plus stem dry weight.

### 2.4. Chlorophyll a, b and Carotenoid Content Measurements

Chl *a*, *b* and carotenoid content were measured from mature leaves from each sapling by using acetone (80%, *v/v*). Total amount of Chl *a*, *b* and carotenoid content were determined as described by Arnon [26].

### 2.5. $CO_2$ Assimilation Rate, Stomatal Conductance and Water-Use Efficiency Measurements

The $CO_2$ assimilation rate (A, µmol m$^{-2}$ s$^{-1}$) and stomatal conductance (g$_s$, mol m$^{-2}$ s$^{-1}$) were determined before harvesting the plants by using a portable Infrared Gas Exchange analyzer system (CIRAS-3 Amesbury, MA, USA). All the measurements were taken around 10:00 to 12:00. During the measurements, the leaf chamber temperature was set at 27 °C, relative humidity was kept at 65%, and the reference $CO_2$ was also adjusted to 400 µmol mol$^{-1}$. Intrinsic water use efficiency (WUE$_i$) was measured as the ratio of net $CO_2$ assimilation rate and stomatal conductance [24].

*2.6. Proline, Soluble Sugar, Total Soluble Protein and Total Phenolic Contents Measurements*

Total phenolic content was estimated in leaf samples according to the protocol of Ainsworth [27] with the help of Folin–Ciocalteu reagent. Proline content was determined using the method described by Bates [28]. Soluble protein was estimated according to the method demonstrated by Bradford [29]. Soluble sugar was measured using the anthrone method as demonstrated by Yemm [30].

*2.7. Malondialdehyde Contents and Electrolyte Leakage (EL%) Measurements*

Malondialdehyde (MDA) contents were quantified (a proxy for lipid peroxidation) according to the protocol of Aravind [31]. Membrane permeability was calculated by estimated leaf electrolyte leakage (EL%) according to the protocol of Nayyar [32]. 0.2 g leaf samples were rinsed with deionized water and put into test tubes with 30 mL of distilled water for 12 h. Initial electric conductivity (ECi) was calculated with an EC meter (Model DJS-1C Model DJS-1C; Shanghai Analytical Instrument Co. Shanghai, China). Then the leaf samples were heated at 100 °C for 20 min, cooled to room temperature, and final electric conductivity (ECf) was measured. EL% was calculated using the following formula:

$$EL\ (\%) = (ECi/ECf) \times 100$$

*2.8. Hydrogen Peroxide ($H_2O_2$) and Superoxide Radical ($O_2{}^-$) Measurements*

To determine the concentration of $H_2O_2$ we followed the protocol of Velikova [33]. Superoxide radicals ($O_2{}^-$) were measured as demonstrated by Bai [34].

*2.9. Antioxidants Enzyme, Superoxide Dismutase (SOD), Peroxidase (POD), Catalase (CAT) and Ascorbate Peroxidase (APX) Measurements*

The SOD concentration was measured by photochemical reduction of NBT (nitroblue tetrazolium) according to the protocol of Bayer [35]. POD concentration was estimated according to Maehly [36]. CAT was determined as described by Knörzer [37]. The reaction mixture was prepared with 200 μL enzyme extract, 1.5 mL of 100 mM potassium phosphate buffer (pH 7.8), 200 μL of 30 mM $H_2O_2$. The absorbance was measured through the quantity of $H_2O_2$ consumed at 240 nm which was decreased by 0.01/min. APX enzyme concentration was measured following the protocol described by Nakano [38].

*2.10. Data Analysis*

The data corresponding to the morphology, physiology and biochemical traits was analyzed using a one-way analysis of variance (ANOVA) where treatment was chosen as fixed effect. Significant differences between treatment means were evaluated using Turkey's honestly significant difference (HSD) post hoc test. All tests and correlations were considered significant at $p < 0.05$ and all the means were represented along with their standard errors (±SE). Analyses were conducted in STATISTICA (Version 12.5 USA).

**3. Results**

*3.1. Effect of Water Deficit and SA on Growth and Biomass Attributes*

Water deficit had a harmful effect on plant growth and dry weight production. Plant height and stem diameter remained similar to CK under MS; however, they decreased significantly under HS (10% and 33%, respectively; Table 1). Leaf, stem and total dry weight decreased significantly under MS and further decreased significantly under HS by 36%, 47%, and 25%, respectively, as compared to the CK. Root dry weight increased by 11% under MS and remained similar under HS (Figure 1). As a result, the R:S ratio increased significantly under MS and HS by 60% and 80%, respectively compared to CK (Table 1).

**Table 1.** Mean (±SE) values of plant height, stem diameter, number of leaves and root:shoot (R:S) ratio measured in water deficit and salicylic acid (SA) application treatments. Data were tested by using a one-way analysis of variance (ANOVA) for treatment effects and differences were considered as significant at $p < 0.05$. Small letters represent significant differences between treatments tested using Tukey's honestly significant difference (HSD) post hoc test.

| Traits | Plant Height (cm) | Stem Diameter (mm) | Number of Leaves | R:S Ratio |
|---|---|---|---|---|
| CK | 50.54 ± 0.61 abc | 5.87 ± 0.17 a | 29.80 ± 1.96 a | 0.50 ± 0.00 d |
| MS | 47.75 ± 0.85 bc | 5.24 ± 0.23 ab | 23.6 ± 1.6 ab | 0.80 ± 0.01 ab |
| HS | 45.4 ± 1.66 c | 3.91 ± 0.16 c | 15.8 ± 1.2 c | 0.90 ± 0.06 a |
| MS + 0.5 | 56.8 ± 1.20 abc | 5.50 ± 0.16 ab | 26.6 ± 1.88 a | 0.71 ± 0.01 bc |
| HS + 0.5 | 51.8 ± 1.06 abc | 4.61 ± 0.27 bc | 16.0 ± 0.79 c | 0.80 ± 0.01 ab |
| MS + 1.0 | 60.4 ± 0.91 a | 6.02 ± 0.15 a | 28.8 ± 1.59 a | 0.59 ± 0.01 cd |
| HS + 1.0 | 58.4 ± 0.68 ab | 5.31 ± 0.20 ab | 19.8 ± 0.2 bc | 0.72 ± 0.03 bc |
| *p*-values | $p = 0.001$ | $p = 0.001$ | $p = 0.001$ | $p < 0.001$ |

CK, control; MS, medium stress; HS, high stress; MS + 0.5, Medium stress + 0.5 mM of SA; HS + 0.5, High stress + 0.5 mM of SA; MS + 1.0, Medium stress +1.0 mM of SA; HS + 1.0, High stress + 1.0 mM of SA.

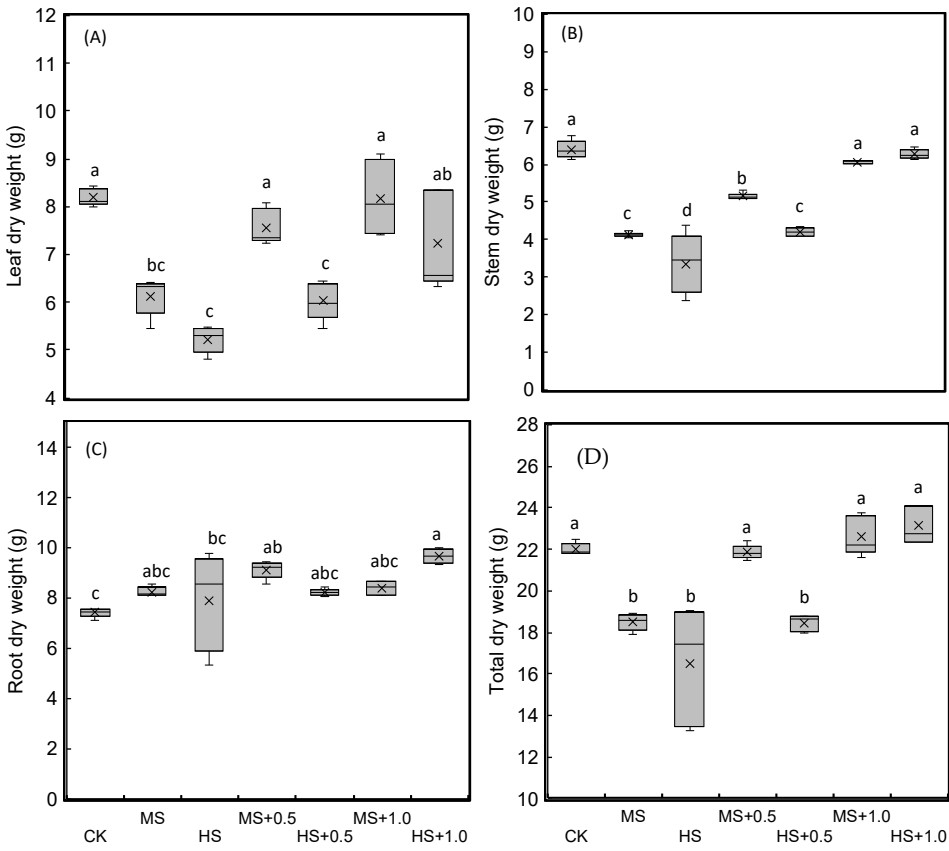

**Figure 1.** Dry weight in (**A**) leaf, (**B**) stem, (**C**) root, (**D**) total dry weight in control (CK), medium stress (MS), high stress (HS), MS + 0.5, HS + 0.5, MS + 1.0 and HS + 1.0, were measured. One-way ANOVA was used to test the treatment effect and tests were considered significant at $p < 0.05$. Lower and upper whiskers represent the minimum and maximum values, and the lower and upper limits of each box are the 25th and 75th quartile. "x" represents the mean value, and the horizontal bar represents the median value for the whole data set. Treatments were compared using Tukey's HSD test and the results are represented with small letters.

The foliar application of SA significantly enhanced growth and dry weight production in *S. cumini* saplings under both MS and HS treatments (Table 1 and Figure 1). The highest increase in plant height and stem diameter was evidenced under HS + 1.0 as compared to

HS (28% and 35%, respectively). Similarly, the highest increase in leaf, stem, root and total dry weight was also observed in plants under HS + 1.0 as compared to HS (56%, 80%, 6%, and 37%, respectively).

### 3.2. Effect of Water Deficit and SA on Chlorophyll a,b and Carotenoid Content

Plants under water deficit conditions revealed a significant reduction in chl *a*, *b* and carotenoid content when compared to CK (Table 2). Under HS, chl *a*, *b*, and carotenoids content significantly decreased by 44%, 40% and 38%, respectively, compared to CK. However, the saplings sprayed with SA were less affected than non-treated saplings. Moreover, the saplings sprayed with 1.0 mM concentration of SA showed the highest value of chl *a*, *b*, and carotenoid content as compared to other treatments. The highest increase in chl *a*, *b* and carotenoid contents was also found in plants under HS + 1.0 as compared to HS (39%, 44% and 37%, respectively).

**Table 2.** Mean (±SE) values of chlorophyll *a*, *b*, carotenoid content, proline, soluble sugar, total phenolic contents and soluble protein measured in water deficit and SA application treatments. Data was tested by using a one-way ANOVA for treatment effects and differences were considered as significant at $p < 0.05$. Small letters represent significant differences between treatments tested using Tukey's HSD post hoc test.

| Traits | Chlorophyll *a* (mg g$^{-1}$ FW) | Chlorophyll *b* (mg g$^{-1}$ FW) | Carotenoids (mg g$^{-1}$ FW) | Proline (μ mol g$^{-1}$ FW) | Soluble Sugar (mg g$^{-1}$ FW) | Total Phenolic Content (mg g$^{-1}$ FW) | Soluble Protein (mg g$^{-1}$ FW) |
|---|---|---|---|---|---|---|---|
| CK | 1.55 ± 0.06 a | 1.41 ± 0.10 a | 0.78 ± 0.03 a | 12.5 ± 0.75 d | 74.3 ± 2.05 c | 1.55 ± 0.05 e | 21.3 ± 0.54 e |
| MS | 1.14 ± 0.02 c | 1.11 ± 0.00 b | 0.63 ± 0.02 b | 16.8 ± 0.32 c | 83.0 ± 0.58 b | 2.10 ± 0.02 d | 25.5 ± 0.30 cd |
| HS | 0.86 ± 0.02 d | 0.84 ± 0.02 c | 0.48 ± 0.01 c | 20.7 ± 0.74 b | 84.3 ± 0.27 b | 2.76 ± 0.03 ab | 27.8 ± 0.29 b |
| MS + 0.5 | 1.29 ± 0.01 b | 1.21 ± 0.00 b | 0.76 ± 0.02 a | 20.7 ± 0.36 b | 85.1 ± 0.58 ab | 2.29 ± 0.07 b | 24.9 ± 0.17 d |
| HS + 0.5 | 1.12 ± 0.02 c | 1.19 ± 0.02 b | 0.60 ± 0.02 b | 25.2 ± 0.46 a | 87.0 ± 0.60 ab | 2.85 ± 0.06 a | 28.2 ± 0.28 ab |
| MS + 1.0 | 1.34 ± 0.01 b | 1.29 ± 0.00 ab | 0.80 ± 0.00 a | 25.6 ± 0.52 a | 86.9 ± 0.65 ab | 2.38 ± 0.01 c | 26.6 ± 0.45 bc |
| HS + 1.0 | 1.20 ± 0.00 bc | 1.21 ± 0.02 b | 0.66 ± 0.01 b | 27.6 ± 0.51 a | 88.6 ± 0.53 a | 2.92 ± 0.02 a | 29.5 ± 0.37 a |
| *p*-value | $p < 0.001$ | $p < 0.001$ | $p < 0.001$ | $p < 0.001$ | $p < 0.001$ | $p < 0.001$ | $p < 0.001$ |

CK, control; MS, medium stress; HS, high stress; MS + 0.5, Medium stress + 0.5 mM of SA; HS + 0.5, High stress + 0.5 mM of SA; MS + 1.0, Medium stress +1.0 mM of SA; HS + 1.0, High stress + 1.0 mM of SA.

### 3.3. Effect of Water Deficit and SA on Proline, Soluble Sugar, Total Phenolic Contents and Soluble Protein

The proline and soluble sugar contents significantly increased under MS and HS, respectively (Table 2). However, the highest concentration was evidenced under HS (65% and 13%) as compared to CK. Moreover, the total phenolic content and soluble protein significantly increased under MS and under HS (phenolic content: 35% and 78%, soluble protein: 19% and 30%). In SA-treated saplings the proline and soluble sugar contents significantly increased compared to non-treated saplings (Table 2) and the highest increase was evidenced under HS + 1.0 mM as compared to HS (33% and 6%, respectively). Similarly, the highest increase in total phenolic content and soluble protein was observed under MS + 1.0 and HS + 1.0 as compared to MS and HS (13%, 6%, 4% and 6%).

### 3.4. Effect of Water Deficit and SA on $CO_2$ Assimilation Rate, Stomatal Conductance and Intrinsic Water Use Efficiency

Water deficit treatments significantly decreased the net $CO_2$ assimilation rate and stomatal conductance in both MS and HS (Figure 2). The highest reduction was found under HS where a reduction of 32% and 16% was observed in stomatal conductance and $CO_2$ assimilation rate, respectively as compared to CK. However, the application of SA significantly improved the $CO_2$ assimilation rate under HS + 1.0. No significant variation was evidenced in stomatal conductance after the application of SA under both MS and HS. Intrinsic $WUE_i$ (as the ratio of $CO_2$ assimilation rate and stomatal conductance) significantly increased in both MS and HS as compared to CK; however, the highest increment was found under HS. No significant increase was evidenced in $WUE_i$ after the foliar application of SA except in saplings under HS + 1.0.

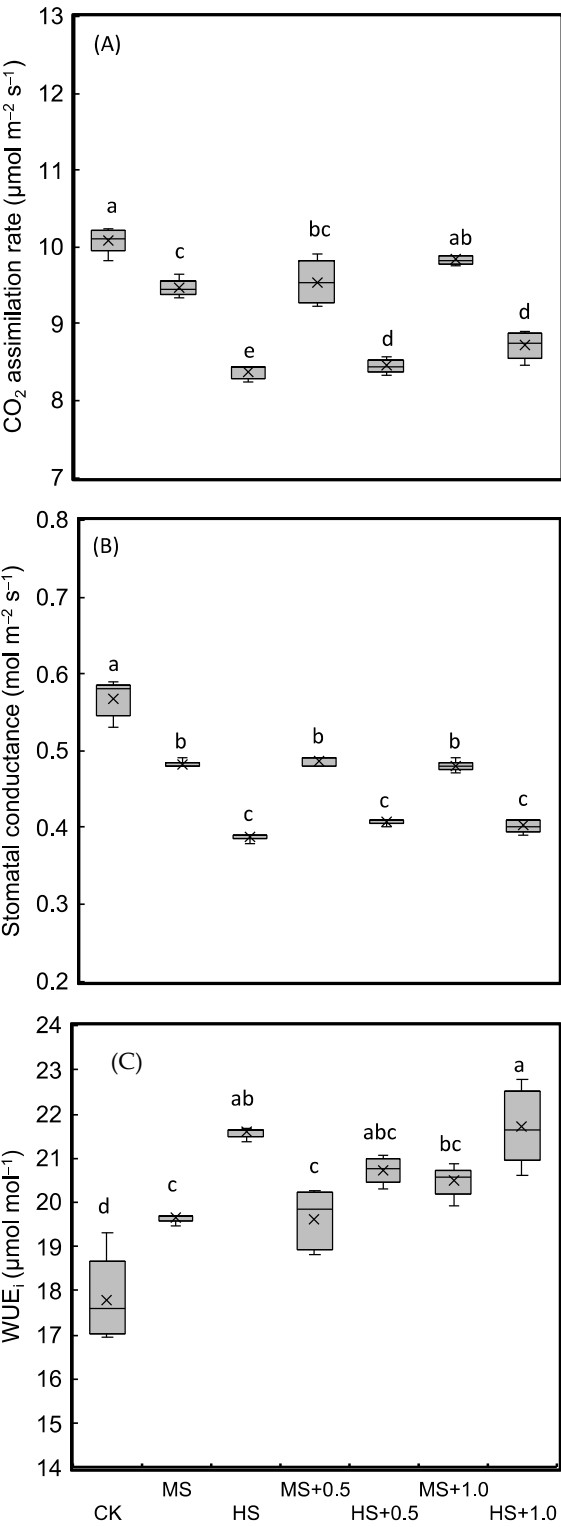

**Figure 2.** Gas exchange parameters (**A**) $CO_2$ assimilation rate, (**B**) stomatal conductance, (**C**) water use efficiency in control (CK), medium stress (MS), high stress (HS), MS + 0.5, HS + 0.5, MS + 1.0 and HS + 1.0, were measured. One-way ANOVA was used to test the treatment effect and tests were considered significant at $p < 0.05$. Lower and upper whiskers represent the minimum and maximum values, and the lower and upper limits of each box are the 25th and 75th quartile. "x" represents the mean value, and the horizontal bar represents the median value for the whole data set. Treatments were compared using Tukey's HSD test and the results are represented with small letters.

### 3.5. Effect of Water Deficit and SA on Lipid Peroxidation (Malondialdehyde, MDA), Hydrogen Peroxide ($H_2O_2$) Superoxide Radical ($O_2^-$) and Electrolyte Leakage (EL%)

Lipid peroxidation was measured in terms of MDA content. The saplings subjected to soil water deficit treatments MS and HS exhibited a significant increase in MDA content by 22% and 49% as compared to CK (Figure 3). Similarly, under both water deficit treatments significantly induced oxidative stress resulted in the generation of ROS such as hydrogen peroxide ($H_2O_2$) and superoxide radicals ($O_2^-$); highest concentrations of $H_2O_2$ and $O_2^-$ were found at HS with a 47% and 41% increase as compared to CK. Moreover, the electrolyte leakage (EL%) significantly increased under MS and HS (19% and 30%) as compared to CK (Figure 3). The foliar application of SA significantly decreased the MDA content as compared with non-treated saplings. Highest decrease was found at MS + 1.0 and HS + 1.0 concentration of SA when compared with HS (22% and 33%). Moreover, the concentration of $H_2O_2$ and $O_2^-$ along with EL% significantly decreased under HS + 1.0 as compared to HS (18%, 12% and 10%, respectively).

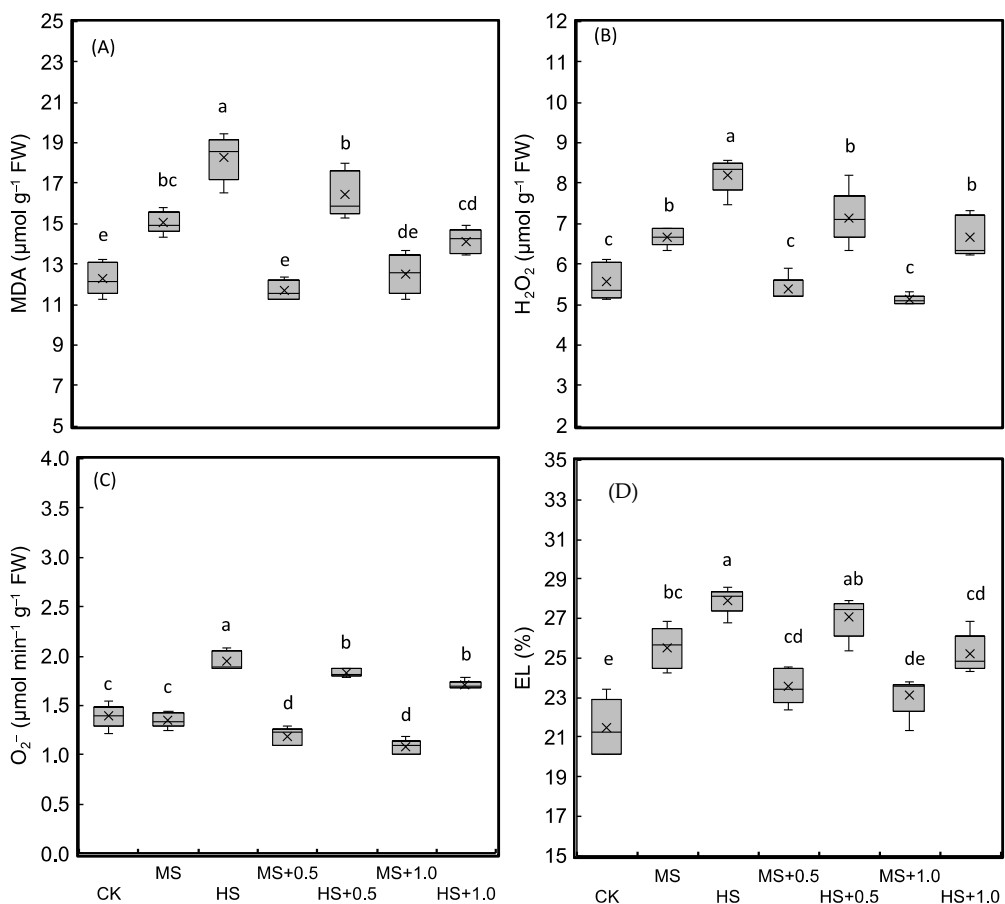

**Figure 3.** Various oxidants (**A**) malondialdehyde (MDA), (**B**) hydrogen peroxide ($H_2O_2$), (**C**) super-oxide radical ($O_2^-$), (**D**) electrolyte leakage (EL%) in control (CK), medium stress (MS), high stress (HS), MS + 0.5, HS + 0.5, MS + 1.0 and HS + 1.0, were measured. One-way ANOVA was used to test the treatment effect and tests were considered significant at *p* < 0.05. Lower and upper whiskers represent the minimum and maximum values, and the lower and upper limits of each box are the 25th and 75th quartile. "x" represents the mean value, and the horizontal bar represents the median value for the whole data set. Treatments were compared using Tukey's HSD test and the results are represented with small letters.

### 3.6. Effect of Water Deficit and SA on Antioxidant Enzymes (SOD, POD, CAT and APX)

After exposure the water deficit treatments, populations significantly increased antioxidant enzyme concentrations of SOD, POD, CAT and APX, however, the highest

concentration was found under HS (78%, 79%, 67%, and 60%) compared to CK (Figure 4). SA-treated plants significantly increased the concentrations of antioxidant enzymes SOD, POD, CAT and APX as compared to non-treated plants. By increasing the concentration of SA, all antioxidant enzymes were increased. Compared with MS and HS, the SOD, POD and CAT concentrations increased by 15%, 19%, 65%, 17%, 20% and 20%, respectively, under MS + 1.0, HS + 1.0 treatments. However, the activity of the APX enzyme remained similar at both concentrations of SA (0.5 and 1.0 mM) under MS and HS, respectively (Figure 4).

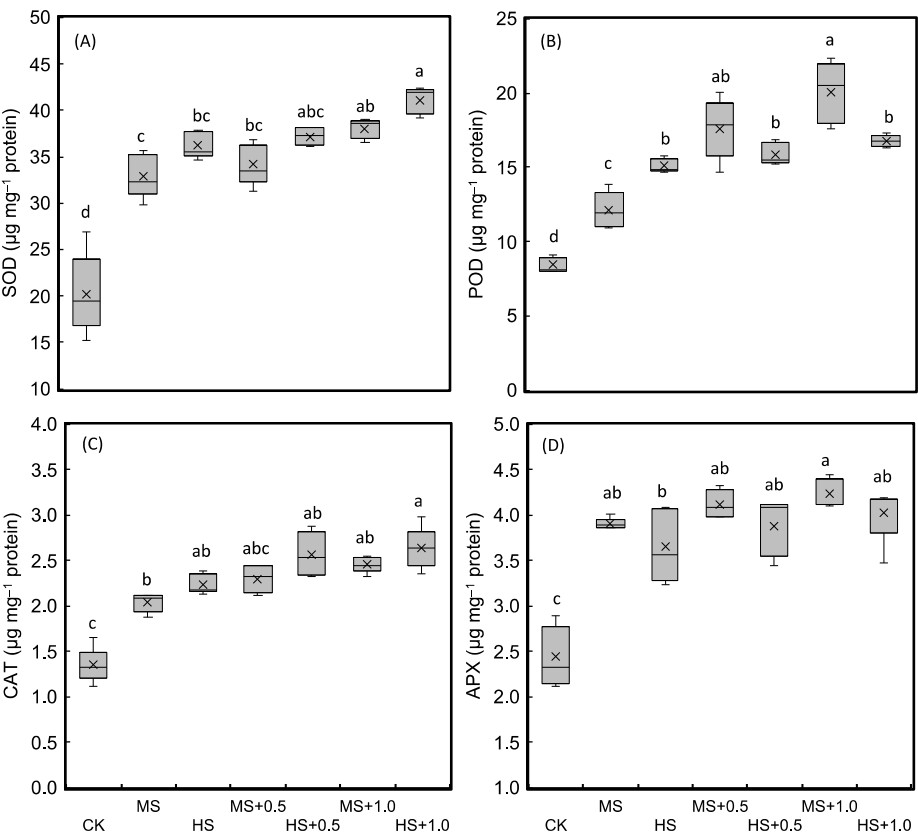

**Figure 4.** Various antioxidant enzyme concentrations for (**A**) superoxide dismutase (SOD), (**B**) peroxidase (POD), (**C**) catalase (CAT), (**D**) ascorbate peroxidase (APX) in control (CK), medium stress (MS), high stress (HS), MS + 0.5, HS + 0.5, MS + 1.0 and HS + 1.0, were measured. One-way ANOVA was used to test the treatment effect and tests were considered significant at *p* < 0.05. Lower and upper whiskers represent the minimum and maximum values, and the lower and upper limits of each box are the 25th and 75th quartile. "x" represents the mean value, and the horizontal bar represents the median value for the whole data set. Treatments were compared using Tukey's HSD test and the results are represented with small letters.

## 4. Discussion

### 4.1. Effect of Water Deficit and SA on Growth and Biomass Attributes

Decrease in plant growth and development is the first and most concerning effect of water stress in plants. In this study, the water stressed saplings of *S. cumini* showed a significant reduction in plant height, stem diameter, number of leaves, leaf dry weight, stem dry weight and total dry weight (Table 1 and Figure 1). These results are in agreement with previous studies [39–41] where decrease in growth and biomass accumulation was observed under water deficit treatments. Studies have shown that reduction in plant growth in the shoot portion of the plants is mostly related to a decrease in meristematic activity which is closely related to a reduction in leaf turgor pressure that negatively affects cell development and growth [42]. However, in the present study, no significant reduction

was observed in root dry weight production under MS or HS (Figure 1). The results suggest that morphological adaption due to altered resource allocation may help *S. cumini* to adapt to water stress conditions. Similar findings have been reported previously in *Ziziphus nummularia* and *Ziziphus jujuba* species where increased root biomass under water stress has been linked to the water stress tolerance strategy in these species [43,44].

SA is an important plant hormone that plays a significant role in enhancing plant tolerance to biotic or abiotic stresses [45]. In the present study, foliar application of SA increased growth rate and biomass accumulation in *S. cumini* saplings under water deficit treatments (Table 1 and Figure 1). Our observation was in agreement with previous studies on *Rosmarinus officinalis* [46], *Portulaca oleracea* [47], *Euclayptus globulus* [19] and *Torreya grandis* [48] where a similar increase in growth and biomass accumulation has been reported in response to SA application under water deficit treatments. Furthermore, the increase in growth attributes after the application of SA has been related to an increase in cell division in the meristematic regions of the plant sapling, thus promoting plant growth and productivity [49].

### 4.2. Effect of Water Deficit and SA on Photosynthetic Pigments and Leaf Gas Exchange Parameters

Stomatal closure is an obvious response of plants to water stress environments, which subsequently decreases transpiration rate as well as concentration of $CO_2$ in the intercellular spaces due to excessive stomatal closure under severe water stress conditions [11]. In the present investigation a significant reduction in both stomatal conductance and $CO_2$ assimilation rate was evidenced under water deficit treatments (Figure 2). Excessive reduction in stomatal conductance has been accepted as a key limiting factor in sustaining photosynthesis and growth under limited supply of water [50]. In *S. cimini* saplings the decrease in $CO_2$ assimilation rate was parallel to stomatal conductance under water deficit treatments. A significant reduction in chl *a*, *b* and carotenoid content was also evidenced under water deficit treatments (Table 2) which refers to an overall decrease in the photosynthetic capacity of the plant under water stress. These results are in line with previous studies of [39,44,47,48] where a similar decrease in stomatal conductance and net $CO_2$ assimilation rate has been reported in *Torreya grandis*, *Portulaca oleracea*, *Conocarpus erectus*, *Ficus benjamina* and *Ziziphus jujuba* plants under water deficit treatments. Studies have demonstrated that plants adjust water loss through transpiration by altering the stomatal aperture that helps maintaining an optimum water potential under water stress [51]. Such adjustment in stomatal aperture under water deficit stress helps maintaining optimal leaf water potential, under severe stress such adjustment may cause a decrease in the $CO_2$ assimilation rate. Many studies have linked the decrease in $CO_2$ assimilation rate and low concentration of $CO_2$ in the intercellular spaces with the excessive reduction in stomatal conductance under severe water stress [44,47]. These conditions create disruptions in the photochemical activity of the photosystem-2 (PS-2) and hinder the electron supply, thus decreasing the overall photosynthesis activity [52]. In this study, the foliar application of SA resulted in a significant increase in stomatal conductance and $CO_2$ assimilation rate under water deficit treatments (Figure 2). Studies have shown that SA is a phytohormone that prevents chlorophyll degradation under stressful environments by decreasing the damage to the photosynthetic apparatus. Such an alteration is induced by the increase in compatible solutes accumulation, the activity of chlorophyll synthesizing enzymes and decrease in the concentration of ROS after SA application [8]. Therefore, the observed increase in chl *a*, *b* and carotenoid content in this study indicated that SA helped to alleviate the harmful effect of water stress [47,48] (Table 2). These results are in agreement with previous study where a positive effect of foliar application of SA has been observed on the impaired photosynthetic processes under water stressed plants in *Portulaca oleracea* [47]. Furthermore, studies have shown that the foliar application of SA enhances the $CO_2$ assimilation rate and growth under stressful environments by increasing the activity and efficiency of carboxylation by the rubisco enzyme under reduced stomatal conductance due to stressful environments [53]. Intrinsic water use efficiency (WUE$_i$, the ratio between

$CO_2$ assimilation rate and stomatal conductance) varied significantly among the treatments with a progressive increase observed under MS and HS (Figure 2). The results are in line with previous studies [24,39] where increase in $WUE_i$ has been reported in plants under limited availability of water. However, $WUE_i$ increased significantly only under HS + 1.0, which was mediated by a significant increase in the $CO_2$ assimilation rate under high stress treatment after foliar application of SA.

### 4.3. Effect of Water Deficit and SA on Proline, Soluble Sugar, Total Phenolic Contents and Soluble Protein

Compatible solutes such as proline are considered an important physiological adaptation in plants to resist water deficit environments and play a significant role in regulating cell turgor, gas exchange and growth attributes under very dry conditions. The role of soluble sugars in enhancing plant tolerance to water stress tolerance has been widely reported in previous studies e.g., [8,15,54,55]. In this study, proline and soluble sugars significantly increased under MS and HS, respectively (Table 2). Similar results have been found in *Quercus*, *Torreya grandis* and *Olea europaea* saplings under water deficit [48,56,57]. Similarly, increase in soluble sugars has also been found in *Portulaca oleracea*, *Populus nigra* and *Eucalyptus globulus* plants under water deficit treatments [19,47,58]. Increase in proline concentration plays a key role in osmotic adjustment and protects the plant cell from ROS damage under water stress [59]. Likewise, studies have shown that increase in the concentration of soluble sugars under water stress is related to the breakdown of polysaccharides such as starch into smaller soluble sugars such as glucose. Furthermore, it has been demonstrated that the increase in soluble sugars plays a significant role in maintaining the osmotic potential and cellular turgor pressure in water deficit environments [60]. In this study, foliar application of SA further increased the concentration of proline and soluble sugars under both water deficit treatments (Table 2). In a previous study it was reported that the increase of proline after foliar application of SA was due to an increase in the activity of the proline-synthesizing enzyme Gamma glutamyl kinase and a decrease in the activity of the proline degrading enzyme proline oxidase [61]. Furthermore, phenolic compounds are also important secondary metabolites that play significant roles in improving plant tolerance to stressful environments [62]. In this study, phenolic compounds increased significantly under water deficit conditions (Table 2). These findings are in line with previous studies showing an increase in phenolic compounds under water deficit treatments [47,63]. Similarly, SA mediated increase in phenolic compounds has been linked to the up-regulation of phenolics-synthesizing enzymes such as phenylalanine ammonia-lyase PAL [64].

### 4.4. Effect of Water Deficit and SA on Oxidants, MDA Contents, EL% and Antioxidants Enzyme

The production of oxidants and antioxidants plays a significant role for the tolerance status of species to different types of abiotic and biotic stresses. Water stress-induced oxidative stress leads to an increase in the concentration of reactive oxygen species (ROS) like hydrogen peroxide ($H_2O_2$), singlet oxygen ($^1O_2$) and superoxide radicals $O_2{}^-$ [65]. In this experiment, the enhancement of $H_2O_2$ and $O_2{}^-$ was observed in *S. cumini* saplings under both MS and HS (Figure 3). These findings are in line with previous studies where an increase of $H_2O_2$ and $O_2{}^-$ has been evidenced in *Torreya grandis* and *Portulaca oleracea* plants under water deficit [47,48]. Different studies demonstrated that increase in production of ROS disrupts the redox balance which ultimately reduces plant productivity under water stress [66,67]. Electrolyte leakage (EL%) is considered an indicator of cell membrane stability and integrity and an increase in EL% as compared to control reflects the degree of damage to the plant under stress conditions [68]. Furthermore, an increase in MDA contents is related to the degree of lipid peroxidation due to oxidative stress [69]. In this experiment, both EL% and MDA content significantly increased in *S. cumini* saplings under MS and HS, respectively (Figure 3). Similar results have been reported in previous studies where an increase in EL% and MDA contents has been reported in plants that were subjected to water deficit [56,63,70]. It has been demonstrated that the increase in EL% is related to the lipid peroxidation of cell membranes and osmotic imbalance [71].

Studies have demonstrated that species having an effective antioxidant defense mechanism showed a better tolerance and adaptability under a water stress environment [48,67,72]. The antioxidant defense mechanism includes the production of enzymes such as SOD, POD, CAT and APX that play an important role in scavenging the overproduction of ROS [62]. In this experiment, a parallel increase in the concentration of both oxidants and antioxidant enzymes was evidenced under MS and HS, respectively (Figure 4). However, the highest concentration of antioxidant enzymes was found under HS. All of these findings are in line with previous studies on other plant species revealing an increased antioxidant enzyme concentration under water deficit treatments [8,47,57,63]. SOD is considered a first line of defense against the overproduction of ROS [73]. Production of SOD plays an important role in neutralizing the $O_2^-$ into $H_2O_2$ which is scavenged further by CAT and POD. CAT is mostly present in the peroxisome and is implicated in balancing out the increased concentration of hydrogen peroxide ($H_2O_2$), while CAT converts $H_2O_2$ to $H_2O$ in the plant cells [74]. In the present study, the foliar application of SA (0.5 and 1.0 mM) significantly decreased the concentration of ROS such as $H_2O_2$ and $O_2^-$ in *S. cumini* saplings under MS and HS, respectively (Figure 3). Similar results have been reported in previous studies showing a reduction in ROS upon foliar application of SA in lemon, verbena and purslane plants [47,66].

Furthermore, in the current study the foliar application of SA significantly increased the concentration of antioxidant enzyme activities like SOD, POD, CAT and APX under MS and HS, respectively (Figure 4). Similar results have been observed in a previous study on *Piriformospora indica* where the application of SA enhanced the antioxidant enzyme activity thereby decreasing the concentration of ROS under various environmental stresses such as drought stress [62]. In *Lippia citriodora* plants, foliar application of SA has been linked to the improvement of the defense mechanisms through regulation of oxidative stress by increased antioxidant enzyme production under water deficit [66]. Many other studies on *Carthamus tinctorius*, *Portulaca oleracea* and *Torreya grandis* also demonstrated an increase of antioxidant enzyme concentrations in plants after foliar application of SA under water deficit treatments, respectively [47,48,75]. Moreover, foliar application of SA resulted in a decrease in EL% and MDA content in *S. cumini* saplings under water stress (Figure 3). These results are in agreement with previous findings where application of SA application resulted in a decrease in EL% and MDA content in *Lippia citriodora*, *Portulaca oleracea*, and *Eucalyptus globulus* plants under water deficit treatments [19,47,66], thus highlighting the protective role of foliar application of SA under water deficit. Therefore, it can be concluded that the foliar application of SA significantly decreased the concentration of $H_2O_2$, $O_2^-$, MDA and EL% which was mediated by the increased production of antioxidant enzymes.

## 5. Conclusions

In this experiment we were able to demonstrate that soil water deficit treatments had a significant negative effect on *S. cumini* saplings as a significant decrease was evidenced in various growth parameters, dry weight production and physiological attributes (stomatal conductance and $CO_2$ assimilation rate). However, increase in growth parameters, dry weight production and increase in compatible solutes (proline and soluble sugar) and secondary metabolites such as total phenolic contents and soluble protein was observed in plants under foliar application of SA. Furthermore, a significant increase in the concentration of antioxidant enzymes such as SOD, POD, CAT and APX was also observed under both water deficit treatments (MS and HS). Therefore, it can be concluded that the foliar application of SA can help alleviating the negative effects of water stress on *S. cumini* thereby reducing the concentration of ROS such as $H_2O_2$, $O_2^-$ and MDA content and EL% in *S. cumini* saplings under water deficit treatments. Moreover, the results also showed that the concentration of 1.0 mM SA was more effective in improving the plant growth and biomass productivity than 0.5 mM SA. However, the experiment was conducted under controlled

conditions, therefore, further experiments are necessary to validate the effectiveness of using the foliar application of SA under field conditions.

**Author Contributions:** Data curation, F.R., O.G.; Formal analysis, Z.Z., F.R.; Supervision, F.R.; Validation, R.M.A., M.M.; Writing—original draft, Z.Z.; Writing—review and editing, F.R., O.G. and M.M. All authors have read and agreed to the published version of the manuscript.

**Funding:** This research received funding from Offices of Research, Innovation and Commercialization (ORIC) Grant no. A/C AO5213, University of Agriculture Faisalabad, Pakistan.

**Acknowledgments:** We are thankful to Muhammad Siddique for providing technical supports during the experiment. We are also thankful to the anonymous reviewers for improving our manuscript.

**Conflicts of Interest:** The authors declare no conflict of interest.

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
