# Peer review of "Salicylic Acid-Induced Morpho-Physiological and Biochemical Changes Triggered Water Deficit Tolerance in Syzygium cumini L. Saplings"

_forests, doi:10.3390/f12040491_

Round 1

Reviewer 1 Report

Minor revision

The paper is interesting, relevant, and well designed. However, it is necessary to make some corrections throughout the text to improve the quality of the presentation. I marked the places in the paper (yellow color) that require checking or making corrections.

Check the beginning of the paper title (It seems that better to use Salicylic Acid-Induced… instead of Salicylic Acid induced…..).

Line 20. R:S ratio -  should be deciphered here.

Lines 135-136. Are the units represented correctly?

Line 144. FC reagent….should be deciphered.

Line 148. I recommend to use 2.7. Lipid peroxidation (MDA content) and …. Instead of 2.7. Lipid peroxidation, MDA contents and ….

Line 156. Аdd a comment before the formula. For example, EL % was calculated using the following formula: …..

Lines 152, 165.  Units of measurement should be indicated in the same manner everywhere, check, please.

Line 188. Use number of leaves instead of no. of leaves…

Line 191-193. Should be checked. The same for other tables.

Line 252. Should be corrected. It seems that the word Effect of ... was missed. The same for sections 3.6, 4.1-4.5. Check it, please.

Also, Check throughout the text the presence/absence of spaces between words, commas, dots...

Author Response

Reviewer 1

Minor revision

The paper is interesting, relevant, and well designed. However, it is necessary to make some corrections throughout the text to improve the quality of the presentation. I marked the places in the paper (yellow color) that require checking or making corrections.

Response: Thank you for appreciating our work.

Check the beginning of the paper title (It seems that better to use Salicylic Acid-Induced… instead of Salicylic Acid induced…..).

Response: The suggestion has been incorporated in the revised version.

Line 20. R:S ratio -  should be deciphered here.

Response: The suggestion has been incorporated in the revised version.

Lines 135-136. Are the units represented correctly?

Response: The units has been corrected in the revised version.

Line 144. FC reagent….should be deciphered.

Response: The suggestion has been incorporated in the revised version.

Line 148. I recommend to use 2.7. Lipid peroxidation (MDA content) and …. Instead of 2.7. Lipid peroxidation, MDA contents and ….

Response: The suggestion has been incorporated in the revised version.

Line 156. Аdd a comment before the formula. For example, EL % was calculated using the following formula: …..

Response: The suggestion has been incorporated in the revised version.

Lines 152, 165.  Units of measurement should be indicated in the same manner everywhere, check, please.

Response: The discrepancy has been removed throughout the manuscript in the revised version.

Line 188. Use number of leaves instead of no. of leaves…

Response: Corrected in the revised version.

Line 191-193. Should be checked. The same for other tables.

Response: The discrepancy has been removed throughout the manuscript in the revised version.

Line 252. Should be corrected. It seems that the word Effect of ... was missed. The same for sections 3.6, 4.1-4.5. Check it, please.

Response: The suggestion has been incorporated throughout the manuscript in the revised version.

Also, Check throughout the text the presence/absence of spaces between words, commas, dots...

Response: All the errors has been removed throughout the manuscript in the revised version.

Reviewer 2 Report

The manuscript deals with a relevant subject to forests. The ms is interesting, well-written and with a high number of studied variables. However, there are a number of areas in the ms which require improvement and clarification, namely on results section, as presented below. Thus, I recommend that the manuscript should be accepted after substantial revision.

Specific comments:

1 - Lines 22-23: authors should reformulate the sentence. In this form, it seems that electrolyte leakage is an oxidant.

2 - Authors should provide information about plant characteristics (height, dry weight, leaf area) at the beginning of experiment, as well about solar radiation intensity in the greenhouse.

3 - Line 135: authors should correct stomatal conductance units.

4 - How authors explain such low values of chlorophyll a/chlorophyll b and chlorophyll/carotenoids ratios?

5 - Figure 2: How authors explain such low differences of net photosynthesis among treatments?

6 - How authors explain such high values of stomatal conductance? The lowest value (a very high value) was around 0.4 mol m-2 s-1 under high water stress. As a result, WUEi values (the highest around 22) were very low. I think there are some problems with these data.

7 - Lines 321-322: I cannot agree with this sentence. In most situations, it is not true. Authors should reformulate the sentence.

8 - The discussion section should be reformulated in order to do connections of variables and not only to do the discussion in an isolated way. The number of subtitles is exaggerated.

Author Response

Reviewer 2

The manuscript deals with a relevant subject to forests. The ms is interesting, well-written and with a high number of studied variables. However, there are a number of areas in the ms which require improvement and clarification, namely on results section, as presented below. Thus, I recommend that the manuscript should be accepted after substantial revision.

Response: Thank you for appreciating our work.

Specific comments:

1 - Lines 22-23: authors should reformulate the sentence. In this form, it seems that electrolyte leakage is an oxidant.

Response: the sentence has been reworked for clarity in the revised version.

2 - Authors should provide information about plant characteristics (height, dry weight, leaf area) at the beginning of experiment, as well about solar radiation intensity in the greenhouse.

Response: Thank you for asking this information that has helped us correcting a serious mistake in the data of plant height. Previous data of plant height was actually in “inches” and wrongly presented in “cm”. Therefore, we have converted that data into “cm” and incorporated in the revised version.

The information on initial plant height and intensity of solar radiation has been added in the revised version. However, we did not measure the initial dry weight and leaf area of the plants used during the experiment.

3 - Line 135: Authors should correct stomatal conductance units.

Response: Units has been corrected throughout the manuscript in the revised version.

4 - How authors explain such low values of chlorophyll a/chlorophyll b and chlorophyll/carotenoids ratios?

Response: The values reported in this study fall within the range that is reported previously in various studies.

  • Shen et al. 2014: Salicylic acid induces physiological and biochemical changes in Torreya grandis cv. Merrillii seedlings under drought stress. Trees. Values range (0.85 to 1.23 mg g-1 FW).
  • Liu et al. 2019: Physiological and proteomic responses of Mulberry trees (Morus alba L.) to combined salt and drought stress. International journal of molecular Sciences. Values’ range (0.1 to 0.7 mg g-1 FW).

  • Liu et al. 2011: Effect of drought on pigments, osmotic adjustment and antioxidant enzymes in six woody plant species in karst habitats of southwestern China. Environmental and experimental botany. Values’ range (1.8 to 5.9 mg g-1 FW).
  • Saheri et al. 2020: Foliar spray of salicylic acid induces physiological and biochemical changes in purslane (Portulaca oleracea L.) under drought stress. Values’ range (0.62 to 0.89 mg g-1 FW).

5 - Figure 2: How authors explain such low differences of net photosynthesis among treatments?

Response: It has been reported that S. cumini peaks its net CO2 assimilation rate at solar radiation around 600 umol m-2 s-1 of PPFD (Qi et al. 2004). In this study, a significant drop was evidenced under MS and HS for both the CO2 assimilation rate and stomatal conductance that resulted in a significant increase in WUEi under water stress. However, after application of SA, although CO2 assimilation rate and stomatal conductance increased, the magnitude of increase was similar in both CO2 assimilation rate and stomatal conductance, therefore WUEi remained similar after SA application.

6 - How authors explain such high values of stomatal conductance? The lowest value (a very high value) was around 0.4 mol m-2 s-1 under high water stress. As a result, WUEi values (the highest around 22) were very low. I think there are some problems with these data.

Response: The reported value of stomatal conductance for S. cumini under desert conditions is around 2.5 mol m-2 sec-1 (Wei et al. 2011). The said experiment was conducted under field conditions and in harsh environments (desert conditions). Therefore, plants in that experiment were subjected to soil water stress as well as high VPD that had negatively affected stomatal conductance.

On the contrary, in our experiment, the experiment was conducted under controlled conditions where plants were subjected to soil water deficit, the VPD remained around 0.7 to 1 KPa, which may explain the higher values of stomatal conductance.

7 - Lines 321-322: I cannot agree with this sentence. In most situations, it is not true. Authors should reformulate the sentence.

Response: The sentence has been modified in the revised version.

8 - The discussion section should be reformulated in order to do connections of variables and not only to do the discussion in an isolated way. The number of subtitles is exaggerated.

Response: Subtitles of the discussion sections have been reduced and the discussion has been reworked to make it more coherent in the revised manuscript.

Round 2

Reviewer 2 Report

I have read the new version of the manuscript and the authors thoughts on all my previous remarks.

I believe that the manuscript can be accepted for publication.